# On the Concept of Resource-Efficiency in NLP

**Luise Dürlich**[*,1,2]   **Evangelia Gogoulou**[*,1,3]   **Joakim Nivre**[1,2]

[1]RISE Research Institutes of Sweden, Department of Computer Science
[2]Uppsala University, Department of Linguistics and Philology
[3]KTH Royal Institute of Technology, Division of Software and Computer Systems

`{luise.durlich, evangelia.gogoulou, joakim.nivre}@ri.se`

## Abstract

Resource-efficiency is a growing concern in the NLP community. But what are the resources we care about and why? How do we measure efficiency in a way that is reliable and relevant? And how do we balance efficiency and other important concerns? Based on a review of the emerging literature on the subject, we discuss different ways of conceptualizing efficiency in terms of product and cost, using a simple case study on fine-tuning and knowledge distillation for illustration. We propose a novel metric of amortized efficiency that is better suited for life-cycle analysis than existing metrics.

## 1   Introduction

Resource-efficiency has recently become a more prominent concern in the NLP community. The Association for Computational Linguistics (ACL) has issued an Efficient NLP Policy Document[1] and most conferences now have a special track devoted to efficient methods in NLP. The major reason for this increased attention to efficiency can be found in the perceived negative effects of scaling NLP models (and AI models more generally) to unprecedented sizes, which increases energy consumption and carbon footprint as well as raises barriers to participation in NLP research for economic reasons (Strubell et al., 2019; Schwartz et al., 2020). These considerations are important and deserve serious attention, but they are not the only reasons to care about resource-efficiency. Traditional concerns like guaranteeing that models can be executed with sufficient speed to enable real-time processing, or with sufficiently low memory footprint to fit on small devices, will continue to be important as well.

Resource-efficiency is however a complex and multifaceted problem. First, there are many relevant types of resources, which interact in complex (and sometimes antagonistic) ways. For example, adding more computational resources may improve time efficiency but increase energy consumption. For some of these resources, obtaining relevant and reliable measurements can also be a challenge, especially if the consumption depends on both software and hardware properties. Furthermore, the life-cycle of a typical NLP model can be divided into different phases, like pre-training, fine-tuning and (long-term) inference, which often have very different resource requirements but nevertheless need to be related to each other in order to obtain a holistic view of total resource consumption. Since one and the same (pre-trained) model can be fine-tuned and deployed in multiple instances, it may also be necessary to amortize the training cost in order to arrive at a fair overall assessment.

To do justice to this complexity, we must resist the temptation to reduce the notion of resource-efficiency to a single metric or equation. Instead, we need to develop a conceptual framework that supports reasoning about the interaction of different resources while taking the different phases of the life-cycle into account. The emerging literature on the subject shows a growing awareness of this need, and there are a number of promising proposals that address parts of the problem. In this paper, we review some of these proposals and discuss issues that arise when trying to define and measure efficiency in relation to NLP models.[2] We specifically address the need for a holistic assessment of efficiency over the entire life-cycle of a model and propose a novel notion of amortized efficiency. All notions and metrics are illustrated in a small case study on fine-tuning and knowledge distillation.

---

[*]Equal contribution to this work.

[1]https://www.aclweb.org/portal/content/efficient-nlp-policy-document

[2]Most of the discussion is relevant also to other branches of AI, although some of the examples and metrics discussed are specific to NLP.

## 2 Related Work

Strubell et al. (2019) were among the first to discuss the increasing resource requirements in NLP. They provide estimates of the energy needed to train a number of popular NLP models, including T2T (Vaswani et al., 2017), ELMo (Peters et al., 2018), BERT (Devlin et al., 2019) and GPT2 (Radford et al., 2019). Based on those estimates, they also estimate the cost in dollars and the $CO_2$ emission associated with model training. In addition to the cost of training a single model, they provide a case study of the additional (much larger) costs involved in hyperparameter tuning and model fine-tuning. Their final recommendations include: (a) Authors should report training time and sensitivity to hyperparameters. (b) Academic researchers need equitable access to computation resources. (c) Researchers should prioritize computationally efficient hardware and algorithms.

Schwartz et al. (2020) note that training costs in AI increased 300,000 times from 2012 to 2017, with costs doubling every few months, and argue that focusing only on the attainment of state-of-the-art accuracy ignores the economic, environmental, or social cost of reaching the reported accuracy. They advocate research on *Green AI* – AI research that is more environmentally friendly and inclusive than traditional research, which they call *Red AI*. Specifically, they propose making *efficiency* a more common evaluation criterion for AI papers alongside accuracy and related measures.

Hershcovich et al. (2022) focus specifically on environmental impact and propose a climate performance model card that can be used with only limited information about experiments and underlying computer hardware. At a minimum authors are asked to report (a) whether the model is publicly available, (b) how much time it takes to train the final model, (c) how much time was spent on all experiments (including hyperparameter search), (d) what the total energy consumption was, and (e) at which location the computations were performed. In addition, authors are encouraged to report on the energy mix at the location and the $CO_2$ emission associated with different phases of model development and use.

Liu et al. (2022) propose a new benchmark for efficient NLP models called ELUE (Efficient Language Understanding Evaluation) based on the concept of Pareto state of the art, which a model is said to achieve if it achieves the best performance at a

given cost level. The cost measures used in ELUE are number of model parameters and number of floating point operations (FLOPs), while performance measures vary depending on the task (sentiment analysis, natural language inference, paraphrase and textual similarity).

Treviso et al. (2022) provide a survey of current research on efficient methods for NLP, using a taxonomy based on different aspects or phases of the model life-cycle: data collection and preprocessing, model design, training (including pre-training and fine-tuning), inference, and model selection. Following Schwartz et al. (2020), they define efficiency as the cost of a model in relation to the results it produces. They observe that cost can be measured along multiple dimensions, such as computational, time-wise or environmental cost, and that using a single cost indicator can be misleading. They also emphasize the importance of separately characterizing different stages of the model life-cycle and acknowledge that properly measuring efficiency remains a challenge.

Dehghani et al. (2022) elaborate on the theme of potentially misleading efficiency characterizations by showing that some of the most commonly used cost indicators – number of model parameters, FLOPs, and throughput (msec/example) – can easily contradict each other when used to compare models and are therefore insufficient as standalone metrics. They again stress the importance of distinguishing training cost from inference cost, and point out that their relative importance may vary depending on context and use case. For example, training efficiency is crucial if a model needs to be retrained often, while inference efficiency may be critical in embedded applications.

## 3 The Concept of Efficiency in NLP

Efficiency is commonly defined as the ratio of useful output to total input:[3]

$$r = \frac{P}{C} \qquad (1)$$

where $P$ is the amount of useful output or results, the *product*, and $C$ is the total *cost* of producing the results, often defined as the amount of resources consumed. A process or system can then be said

---

[3]Historically, the technical concept of efficiency arose in engineering in the nineteenth century, in the analysis of engine performance (thermodynamic efficiency); it was subsequently adopted in economy and social science by Vilfredo Pareto and others (Mitcham, 1994).

to reach maximum efficiency if a specific desired result is obtained with the minimal possible amount of resources, or if the maximum amount of results is obtained from a given resource. More generally, maximum efficiency holds when it is not possible to increase the product without increasing the cost, nor reduce the cost without reducing the product.

In order to apply this concept of efficiency to NLP, we first have to decide what counts as useful output or results – the product $P$ in Equation 1. We then need to figure out how to measure the cost $C$ in terms of resources consumed. Finally, we need to come up with relevant ways of relating $P$ to $C$ in different contexts of research, development and deployment, as well as aggregating the results into a life-cycle analysis. We will begin by discussing the last question, because it has a bearing on how we approach the other two.

### 3.1 The Life-Cycle of an NLP Model

It is natural to divide the life-span of an NLP model into two phases: *development* and *deployment*. In the development phase, the model is created, optimized and validated for use. In the deployment phase, it is being used to process new text data in one or more applications. The development phase of an NLP model today typically includes several stages of training, some or all of which may be repeated multiple times in order to optimize various hyperparameters, as well as validation on held-out data to estimate model performance. The deployment phase is more homogeneous in that it mainly consists in using the model for inference on new data, although this may be interrupted by brief development phases to keep the model up to date.

As researchers, we naturally tend to focus on the development of new models and many models developed in a research context may never enter the deployment phase at all. Since the development phase is typically also more computationally intensive than the deployment phase, it is therefore not surprising that early papers concerned with the increasing energy consumption of NLP research, such as Strubell et al. (2019) and Schwartz et al. (2020), mainly focused on the development phase. Nevertheless, for models that are actually put to use in large-scale applications, resources consumed during the deployment phase may in the long run be much more important, and efficiency in the deployment phase is therefore an equally valid concern. This is also the focus of the recently proposed evaluation framework ELUE (Liu et al., 2022).

As will be discussed in the following sections, some proposed efficiency metrics are better suited for one of the two phases, although they can often be adapted to the other phase as well. However, the question is whether there is also a need for metrics that capture the combined resource usage at development and deployment, and how such metrics can be constructed. One reason for being interested in combined metrics is that there may be trade-offs between resources spent during development and deployment, respectively, so that spending more resources in development may lead to more efficient deployment (or vice versa). To arrive at a more holistic assessment of efficiency, we need to define efficiency metrics for deployment that also incorporate development costs. Before we propose such a metric, we need to discuss how to conceptualize products and costs of NLP models.

### 3.2 The Products of an NLP Model

What is the output that we want to produce at the lowest possible cost in NLP? Is it simply a model capable of processing natural language (as input or output or both)? Is it the performance of such a model on one or more NLP tasks? Or is it the actual output of such a model when processing natural language at a certain performance level? All of these answers are potentially relevant, and have been considered in the literature, but they give rise to different notions of efficiency and require different metrics and measurement procedures.

Regarding the model itself as the product is of limited interest in most circumstances, since it does not take performance into account and only makes sense for the development phase. It is therefore more common to take model performance, as measured on some standard benchmark, as a relevant product quantity, which can be plotted as a function of some relevant cost to obtain a so-called Pareto front (with corresponding concepts of Pareto improvement and Pareto state of the art), as illustrated in Figure 1, reproduced from Liu et al. (2022).

One advantage of the product-as-performance model is that it can be applied to the deployment phase as well as the development phase, although the cost measurements are different in the two cases. For the development phase, we want to measure the *total* cost incurred to produce a model with a given performance, which depends on a multitude of factors, such as the size of the model, the num-

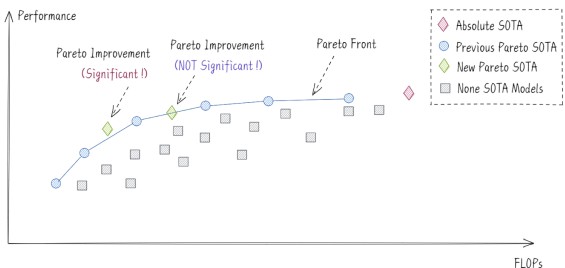

Figure 1: Pareto front with model performance as the product and cost measured in FLOPs (Liu et al., 2022).

ber of hyperparameters that need to be tuned, and the data efficiency of the learning algorithm. For the deployment phase, we instead focus on the *average* cost of processing a typical input instance, such as a natural language sentence or a text document, independently of the development cost of the model. Separating the two phases in this way is perfectly adequate in many circumstances, but the fact that we measure total cost in one case and average cost in the other makes it impossible to combine the measurements into a global life-cycle analysis. To overcome this limitation, we need a notion of product that is not defined (only) in terms of model performance but also considers the actual output produced by a model.

If we take the product to be the amount of data processed by a model in the deployment phase, then we can integrate the development cost in the efficiency metric as a debt that is amortized during deployment. Under this model, the average cost of processing an input instance is not constant but decreases over the life-time of a model, which allows us to capture possible trade-offs between development and deployment costs. For example, it may sometimes be worth investing more resources into the development phase if this leads to a lower development cost in the long run. Moreover, this model allows us to reason about how long a model needs to be in use to "break even" in this respect.

An important argument against the product-as-output model is that it is trivial (but uninteresting) to produce a maximally efficient model that produces random output. It thus seems that a relevant life-cycle analysis requires us to incorporate both model performance and model output into the notion of product. There are two obvious ways to do this, each with its own advantages and drawbacks. The first is to stipulate a minimum performance level that a model must reach to be considered valid and to treat all models reaching this threshold as ceteris paribus equivalent. The second way is to use the performance level as a weighting function when calculating the product of a model. We will stick to the first and simpler approach in our case study later, but first we need to discuss the other quantity in the efficiency equation – the cost.

### 3.3 The Costs of an NLP Model

Schwartz et al. (2020) propose the following formula for estimating the computational cost of producing a result $R$:

$$Cost(R) \propto E \cdot D \cdot H \qquad (2)$$

where $E$ is the cost of executing the model on a single example, $D$ is the size of the training set (which controls how many times the model is executed during a training run), and $H$ is the number of hyperparameter experiments (which controls how many times the model is trained during model development). How can we understand this in the light of the previous discussion?

First, it should be noted that this is not an exact equality. The claim is only that the cost is proportional to the product of factors on the right hand side, but the exact cost may depend on other factors that may be hard to control. Depending on what type of cost is considered – a question that we will return to below – the estimate may be more or less exact. Second, the notion of a *result* is not really specified, but seems to correspond to our notion of *product* and is therefore open to the same variable interpretations as discussed in the previous section. Third, as stated above, the formula applies only to the development phase, where the result/product is naturally understood as the performance of the final model. To clarify this, we replace $R$ (for result) with $P_P$ (for product-as-performance) and add the subscript $T$ (for training) to the factors $E$ and $D$:

$$DevCost(P_P) \propto E_T \cdot D_T \cdot H \qquad (3)$$

Schwartz et al. (2020) go on to observe that a formula appropriate for inference during the deployment phase can be obtained by simply removing the factors $D$ and $H$ (and, in our new notation, changing $E_T$ to $E_I$ since the cost of processing a single input instance is typically not the same at training and inference time):

$$DepCost(P_P) \propto E_I \qquad (4)$$

This corresponds to the product-as-performance model for the deployment phase discussed in the previous section, based on the average cost of processing a typical input instance, and has the same limitations. It ignores the quantity of data processed by a model, and it is insensitive to the initial investment in terms of development cost. To overcome the first limitation, we can add back the factor $D$, now representing the amount of data processed during deployment (instead of the amount of training data), and replace product-as-performance ($P_P$) by product-as-output ($P_O$):

$$DepCost(P_O) \propto E_I \cdot D_I \qquad (5)$$

To overcome the second limitation, we have to add the development cost to the equation:

$$DepCost(P_O) \propto E_T \cdot D_T \cdot H + E_I \cdot D_I \quad (6)$$

This allows us to quantify the product and cost as they develop over the lifetime of a model, and this is what we propose to call *amortized* efficiency based on total deployment cost, treating development cost as a debt that is amortized during the deployment phase. Our notion of amortized efficiency is inspired by the notion of amortized analysis from complexity theory (Tarjan, 1985), which averages costs over a sequence of operations. Here we instead average costs over different life-cycle phases.

As already noted, the product-as-output view is only meaningful if we also take model performance into account, either by stipulating a threshold of minimal acceptable performance or by using performance as a weight function when calculating the output produced by the model. Note, however, that we can also use the notion of total deployment cost to compare the Pareto efficiency of different models at different points of time (under a product-as-performance model) by computing average deployment cost in a way that is sensitive to development cost and lifetime usage of a model.

The discussion so far has focused on how to understand the notion of efficiency in NLP by relating different notions of *product* to an abstract notion of *cost* incurred over the different phases of lifetime of a model. However, as noted in the introduction, this abstract notion of cost can be instantiated in many different ways, often in terms of a specific resource being consumed, and it may be more or less straightforward to obtain precise measures of the resource consumption. Before illustrating the different efficiency metrics with some real data, we will therefore discuss costs and resources that have been prominent in the recent literature and motivate the selection of costs included in our case study.

**Time and Space** The classical notion of efficient computation from complexity theory is based on the resources of *time* and *space*. Measuring cost in terms of time and space (or memory) is important for time-critical applications and/or memory-constrained settings, but in this context we are more interested in execution time and memory consumption than in asymptotic time and space complexity. For this reason, execution time remains one of the most often reported cost measures in the literature, even though it can be hard to compare across experimental settings because it is influenced by factors such as the underlying hardware, other jobs running on the same machine, and the number of cores used (Schwartz et al., 2020). We include execution time as one of the measured costs in our case study.

**Power and $CO_2$** Electrical power consumption and the ensuing $CO_2$ emission are costs that have been highlighted in the recent literature on resource-efficient NLP and AI. For example, Strubell et al. (2019) estimate the total power consumption for training NLP models based on available information about total training time, average power draw of different hardware components (GPUs, CPUs, main memory), and average power usage effectiveness (PUE) for data centers. They also discuss the corresponding $CO_2$ emission based on information about average $CO_2$ produced for power consumed in different countries and for different cloud services. Hershcovich et al. (2022) propose that climate performance model cards for NLP models should minimally include information about total energy consumption and location for the computation, ideally also information about the energy mix at the location and the $CO_2$ emission associated with different phases of model development and use. Against this, Schwartz et al. (2020) observe that, while both power consumption and carbon emission are highly relevant costs, they are difficult to compare across settings because they depend on hardware and local electricity infrastructure in a way that may vary over time even at the same location. In our case study, we include measurements of power consumption, but not carbon emission.

**Abstract Cost Measures** Given the practical difficulties to obtain exact and comparable measure-

ments of relevant costs like time, power consumption, and carbon emission, several researchers have advocated more abstract cost measures, which are easier to obtain and compare across settings while being sufficiently correlated with other costs that we care about. One such measure is model size, often expressed as number of parameters, which is independent of underlying hardware but correlates with memory consumption. However, as observed by Schwartz et al. (2020), since different models and algorithms make different use of their parameters, model size is not always strongly correlated with costs like execution time, power consumption, and carbon emission. They therefore advocate number of floating point operations (FLOPs) as the best abstract cost measure, arguing that it has the following advantages compared to other measures: (a) it directly computes the amount of work done by the running machine when executing a specific instance of a model and is thus tied to the amount of energy consumed; (b) it is agnostic to the hardware on which the model is run, which facilitates fair comparison between different approaches; (c) unlike asymptotic time complexity, it also considers the amount of work done at each time step. They acknowledge that it also has limitations, such as ignoring memory consumption and model implementation. Using FLOPs to measure computation cost has emerged as perhaps the most popular approach in the community, and it has been shown empirically to correlate well with energy consumption (Axberg, 2022); we therefore include it in our case study.

**Data**  The amount of data (labeled or unlabeled) needed to train a given model and/or reach a certain performance is a relevant cost measure for several reasons. In AI in general, if we can make models and algorithms more data-efficient, then they will ceteris paribus be more time- and energy-efficient. In NLP specifically, it will in addition benefit low-resource languages, for which both data and computation are scarce resources.

In conclusion, no single cost metric captures all we care about, and any single metric can therefore be misleading on its own. In our case study, we show how different cost metrics can be combined with different notions of product to analyze resource-efficiency for NLP models. We include three of the most important metrics: execution time, power consumption, and FLOPs.

# 4  Case Study

To illustrate the different conceptualizations of resource-efficiency discussed in previous sections, we present a case study on developing and deploying a language model for a specific NLP task using different combinations of fine-tuning and knowledge distillation. The point of the study is not to advance the state of the art in resource-efficient NLP, but to show how different conceptualizations support the comparison of models of different sizes, at different performance levels, and with different development and deployment costs.

## 4.1  Overall Experimental Design

We apply the Swedish pre-trained language model KB-BERT (Malmsten et al., 2020) to Named Entity Recognition (NER), using data from SUCX 3.0 (Språkbanken, 2022) for fine-tuning and evaluation. We consider three scenarios:

- **Fine-tuning (FT):** The standard fine-tuning approach is followed, with a linear layer added on top of KB-BERT. The model is trained on the SUCX 3.0 training set until the validation loss no longer decreases for up to 10 epochs.

- **Task-specific distillation (TS):** We distill the fine-tuned KB-BERT model to a 6-layer BERT student model. The student model is initialized with the 6 lower layers of the teacher and then trained on the SUCX 3.0 training set using the teacher predictions on this set as ground truth.

- **Task-agnostic distillation (TA):** We distill KB-BERT to a 6-layer BERT student model using the task-agnostic distillation objective proposed by Sanh et al. (2020). Following their approach, we initialize the student with every other layer of the teacher and train on deduplicated Swedish Wikipedia data by averaging three kinds of losses for masked language modelling, knowledge distillation and cosine-distance between student and teacher hidden states. The student model is subsequently fine-tuned on the SUCX 3.0 training set with the method used in the FT experiment.

All three fine-tuned models are evaluated on the SUCX 3.0 test set. Model performance is measured using the F1 score, which is the standard evaluation metric for NER, and model output in number of

| | Distillation Stage | | | Fine-Tuning Stage | | | Evaluation Stage | | | F1 |
|---|---|---|---|---|---|---|---|---|---|---|
| | Time | Power | FLOPs | Time | Power | FLOPs | Time | Power | FLOPs | |
| FT | – | – | – | 0:35:17 | 141.1 | $2.48 \times 10^{16}$ | 0:01:32 | 5.2 | $2.59 \times 10^{15}$ | 87.3 |
| TS | 0:18:30 | 77.1 | $1.64 \times 10^{16}$ | 0:35:17 | 141.1 | $2.48 \times 10^{16}$ | 0:01:09 | 3.1 | $1.71 \times 10^{15}$ | 84.9 |
| TA | 13:06:59 | 6848.9 | $3.65 \times 10^{17}$ | 0:18:53 | 74.4 | $1.69 \times 10^{16}$ | 0:01:15 | 3.3 | $1.71 \times 10^{15}$ | 77.6 |

Table 1: Performance (F1) and cost measurements (Time: hh:mm:ss, Power: Wh, FLOPs) for different stages (Distillation, Fine-tuning, Evaluation) and different development scenarios (Fine-tuning: FT, Task-specific distillation: TS, Task-agnostic distillation: TA).

tokens. We measure three different types of cost during development and deployment: execution time, power consumption and FLOPs. Based on these basic measures, we derive different efficiency metrics for model comparison, as discussed in Section 4.4.

### 4.2 Setup Details

The TextBrewer framework (Yang et al., 2020) is used for the distillation experiments, while the Huggingface Transformers[4] library is used for fine-tuning and inference. More information on hyper-parameters and data set sizes can be found in Appendix A. All experiments are executed on an Nvidia DGX-1 server with 8 Tesla V100 SXM2 32GB. In order to get measurements under realistic conditions, we run different stages in parallel on different GPUs, while blocking other processes from the system to avoid external interference. Each experimental stage is repeated 3 times and measurements of execution time and power consumption are averaged.[5]

The different cost types are measured as follows:

- **Execution time:** We average the duration of the individual Python jobs for each experimental stage.

- **Power consumption:** We measure power consumption for all 4 PSUs of the server as well as individual GPU power consumption, following Gustafsson et al. (2018). Based on snapshots of measured effect at individual points in time, we calculate the area under the curve to get the power consumption in Wh. Since we run the task-agnostic distillation using distributed data parallelism on two

GPUs, we sum the consumption of both GPUs for each TA run.

- **FLOPs:** We estimate the number of FLOPs required for each stage using the estimation formulas proposed by Kaplan et al. (2020), for training (7) and inference (8):

$$\text{FLOP}_T = 6 \cdot n \cdot N \cdot S \cdot B \qquad (7)$$

$$\text{FLOP}_I = 2 \cdot n \cdot N \cdot S \cdot B \qquad (8)$$

where $n$ is the sequence length, $N$ is the number of model parameters, $S$ is the number of training/inference steps, and $B$ is the batch size. The cost for fine-tuning a model is given by $\text{FLOP}_T$, while the evaluation cost is $\text{FLOP}_I$. For distillation, we need to sum $\text{FLOP}_T$ for the student model and $\text{FLOP}_I$ for the teacher model (whose predictions are used to train the student model).

### 4.3 Basic Results

Table 1 shows basic measurements of performance and costs for different scenarios and stages. We see that the fine-tuned KB-BERT model (FT) reaches an F1 score of 87.3; task-specific distillation to a smaller model (TS) gives a score of 84.9, while fine-tuning after task-agnostic distillation (TA) only reaches 77.6 in this experiment. When comparing costs, we see that task-agnostic distillation is by far the most expensive stage. Compared to task-specific distillation, the execution time is more than 40 times longer, the power consumption almost 100 times greater, and the number of FLOPs more than 20 times greater. Although the fine-tuning costs are smaller for the distilled TA model, the reduction is only about 50% for execution time and power consumption and about 30% for FLOPs.

We also investigate whether power consumption can be predicted from the number of FLOPs, as this is a common argument in the literature for preferring the simpler FLOPs calculations over the more

---

[4]https://huggingface.co/docs/transformers/index

[5]Since we repeat stages 3 times for every model instance, task-specific distillation, fine-tuning of the distilled model, and evaluation of FT are repeated 9 times, while evaluation of TS and TA is repeated 27 times.

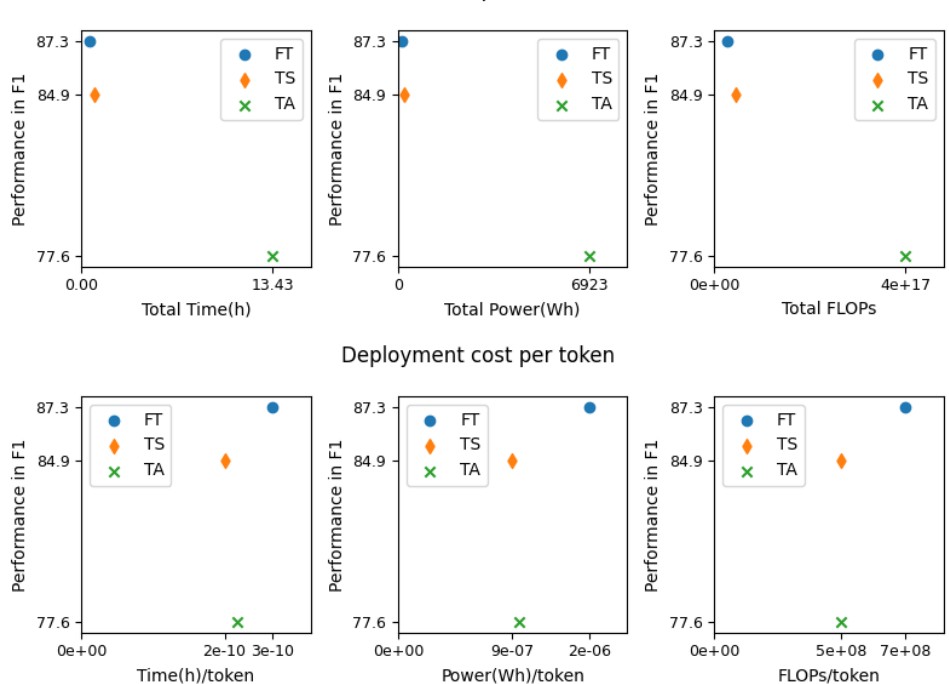

Figure 2: Pareto efficiency for the development phase (top) and the deployment phase (down) based on three different cost measures: execution time (left), power consumption (center), and FLOPs (right).

involved measurements of actual power consumption. We find an extremely strong and significant linear correlation between the two costs (Pearson $r = 0.997$, $p \approx 0$). Our experiments thus corroborate earlier claims that FLOPs is a convenient cost measure that correlates well with power consumption (Schwartz et al., 2020; Axberg, 2022). However, it is worth noting that the GPU power consumption, which is reported in Table 1 and which can thus be estimated from the FLOPs count, is only 71.7% of the total power consumption of the server including all 4 PSUs.

### 4.4 Measuring and Comparing Efficiency

So how do our three models compare with respect to resource-efficiency? The answer is that this depends on what concept of efficiency we apply and which part of the life-cycle we consider. Figure 2 plots product-as-performance as a function of cost separately for the development phase and the deployment phase, corresponding to Equations (3) and (4), which allows us to compare Pareto efficiency. Considering only the development phase, the FT model is clearly optimal, since it has both the highest performance and the lowest cost of all models. Considering instead the deployment phase, the FT model still has the best performance, but the

other two models have lower (average) inference cost. The TA model is still suboptimal, since it gives lower performance at the same cost as the TS model.[6] However, FT and TS are both optimal with respect to Pareto efficiency, since they are both at the Pareto front given the data we have so far (meaning that neither is outperformed by a model at the same cost level nor has higher deployment cost than any model at the same performance level). In order to choose between them, we therefore have to judge whether a 2.4 point improvement in F1 score in the long run is worth the increase in execution time and power consumption, which in this case amounts to 0.077 nano-seconds and 0.607 micro-watts per token, respectively.

For a more holistic perspective on life-time efficiency, we can switch to a product-as-output model and plot deployment efficiency as a function of both the initial development cost and the average inference cost for processing new data over lifetime, corresponding to Equation (6) and our newly proposed notion of amortized efficiency. This is depicted in Figure 3, which compares the FT and

---

[6]It is worth noting, however, that the TA model can be fine-tuned for any number of specific tasks, which could make it competitive in a more complex scenario where we can distribute the initial distillation cost over a large number of fine-tuned models.

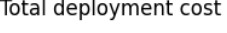

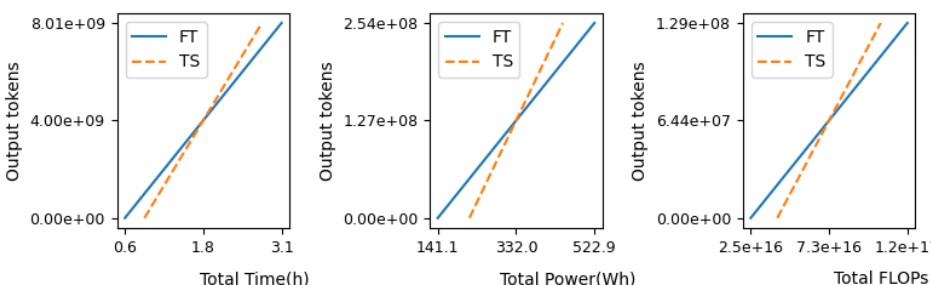

Figure 3: Amortized efficiency of the deployment phase over lifetime, based on three different cost measures: execution time (left), power consumption (center), and FLOPs (right).

TS model (disregarding the suboptimal TA model). We see that, although the FT model has an initial advantage because it has not incurred the cost for distillation, the TS model eventually catches up and becomes more time-efficient after processing about 4B tokens and more energy-efficient after processing about 127M tokens. It is however important to keep in mind that this comparison does not take performance into account, so we again need to decide what increase in cost we are willing to pay for a given improvement in performance, although the increase in this case is sensitive to the expected lifetime of the models. Alternatively, as mentioned earlier, we could weight the output by performance level, which in this case would mean that the TS model would take longer to catch up with the FT model.

Needless to say, it is often hard to estimate in advance how long a model will be in use after it has been deployed, and many models explored in a research context may never be deployed at all (over and above the evaluation phase). In this sense, the notion of life-time efficiency admittedly often remains hypothetical. However, with the increasing deployment of NLP models in real applications, we believe that this perspective on resource-efficiency will become more important.

## 5 Conclusion

In this paper, we have discussed the concept of resource-efficiency in NLP, arguing that it cannot be reduced to a single definition and that we need a richer conceptual framework to reason about different aspects of efficiency. As a complement to the established notion of Pareto efficiency, which separates development and deployment under a product-as-performance model, we have proposed

the notion of amortized efficiency, which enables a life-cycle analysis including both development and deployment under a product-as-output model. We have illustrated both notions in a simple case study, which we hope can serve as inspiration for further discussions of resource-efficiency in NLP. Future work should investigate more sophisticated ways of incorporating performance level into the notion of amortized efficiency.

## Acknowledgments

We would like to thank Jonas Gustafsson and Stefan Alatalo from the ICE data center at RISE for their help with the experimental setup of the case study. Our sincere gratitude goes also to Petter Kyösti and Amaru Cuba Gyllensten for their insightful comments during the development of this work. Finally, we wish to thank the conference reviewers for their constructive feedback.

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

# A  Experimental Details

## A.1  Data Sets

The SUCX 3.0 dataset (simple_lower_mix version)[7] is used for fine-tuning, task-specific distillation and evaluation. The dataset splits are are the following: 43126 examples in the training set, 10772 in the validation set and 13504 examples in the test set.

For task-agnostic distillation, we are using a deduplicated version of Swedish Wikipedia, with the following dataset split: 2,552,479 sentences in the training set and 25,783 sentences in the validation set.

## A.2  Models and Hyperparameters

The base model in our experiments is KB-BERT-cased.[8] The hyperparameters used for fine-tuning and distillation are presented in Table 2. In the fine-tuning experiments, early stopping is used and the best performing model in the validation set is saved. The task-agnostic distillation experiments are performed on two GPUs, using the distributed

---

[7]https://huggingface.co/datasets/KBLab/sucx3_ner
[8]https://huggingface.co/KB/bert-base-swedish-cased

data parallel functionality of pytorch, while gradient accumulation steps are set to 2.

| | FT | TS | TA | Eval |
|---|---|---|---|---|
| Batch size | 32 | 32 | 8 | 32 |
| Training epochs | 10 | 2 | 0.75 | – |
| Sequence length | 256 | 256 | 256 | 256 |
| Learning rate | 0.00003 | 0.00005 | 0.0001 | – |
| Warm-up steps | 404 | 260 | 3750 | – |

Table 2: Hyperparameters for FT, TS, TA and Eval.