# OpenReview forum: "On the Concept of Resource-Efficiency in NLP"
_NoDaLiDa/2023/Conference — NoDaLiDa 2023_

### Official Review · Reviewer_mPSs · 2023-03-09
**Well written analysis of a highly relevant issue**

**Rating:** 9
**Confidence:** 5

**Review:**

I really liked this paper, which proposes a unified framework to evaluate NLP systems not only along their performance on benchmarks, but also on their efficiency. The paper starts with a series of definitions, showing how putting order in this field is not a trivial task. A measure of efficiency appliied to NLP models is then proposed and discussed. Finally, a case study is analysed, where an existing model is tested under the novel framework.
The authors go into details about how to measure several (often constrasting) aspects of the efficiency of NPL systems. However, I think that the most important contribution is the high-level view of the problem, and the proposal of a framework of evaluation, whose individual steps (e.g. specific metrics) could change in the future and adapted to the evolving NLP scenario.
This paper talks about an important problem in clear and methodologically sound terms, and it will be a great contribution to the conference.

**Paper Type:**

Long paper

---

### Official Review · Reviewer_sLdQ · 2023-03-09
**Practical overview of resource efficiency in NLP**

**Rating:** 7
**Confidence:** 3

**Review:**

This paper is both a review of resource efficiency in NLP and a case study on this topic (including a novel metric). The authors discuss in detail which different types of resource costs and metrics can/should be taken into account, and propose to look at both the development and deployment costs in terms of flops, power consumption and time. The paper includes a case study on NER for Swedish, and they show that in most cases simple fine-tuning is preferable over distillation approaches (especially for smaller deployment sizes and with an eye on F1 scores).

In general, the review seems comprehensive (but I'm no expert in this area), and the case study seems informative and concrete enough to build upon. The total amount of new things that can be learned from this paper might be small, but I do see its value in providing suggestions for how to improve.

The one practical difficulty for implementation is that the deployment amount (x-axis figure 3), is in practice very low (and or unknown) for research-based models. Of course we can plot the same graphs as figure 3, but it unclear what to aim for and how to draw conclusions.

Other notes:
- The wrong style file seems to be used (ACL?)
- It should be noted that TS can be re-used as opposed to FT and TS (in the discussion of 4.4 it is regarded as clearly suboptimal and is left out)
- altough the paper is well written, I didnt like that a lot of sections and with; but before we can do X, we need Y. (It breaks the flow in my opinion)

**Paper Type:**

Long paper

---

### Official Review · Reviewer_e423 · 2023-03-10
**Interesting perspective, clear and to the point**

**Rating:** 10
**Confidence:** 5

**Review:**

The paper revisits the notion of resource efficiency in NLP development and deployment, by considering both stages of the life cycle in an amortized cost that allows reasoning about the number of deployment instances at which a particular training cost is worth the performance gap it achieves as compared to others.

The paper starts with a rather comprehensive and informative survey of discussions of efficient NLP, followed by a theoretical background on the concept of efficiency from engineering/economy and how to decompose it to product, cost and aggregating them. This is a principled and clear argumentation that makes perfect sense but I had not seen laid out so well before. The case study nicely demonstrates the applicability of the concept to facilitate choosing a model in an informed way.

In general, the theoretical part seems to be applicable to ML in general and not just NLP. It would be good to clarify that our to highlight which part of the argumentation is NLP-specific.

It might be with briefly explaining the meaning of amortizing a debt, since readers might not be familiar with the concept even though for some people it is everyday financial talk.

In the case study, it is not clear what is meant by a "6-layer BERT student model". Is it a randomly initialized model with the BERT architecture or a pre-trained model? If the latter, is it a Swedish one?

About the conclusion that both FS and TS are Pareto optimal, it might be worth explaining what being at the "same" cost level would mean. Since both cost and performance are continuous measures, how can they be exactly the same?

Finally, why not consider a combined measure of output and performance, e.g., by quantifying the number of correct predictions over time?

**Paper Type:**

Long paper

---

### Decision · Program_Chairs · 2023-03-17

Accept